# Molecular Mechanisms of the Anti-Cancer Effects of Isothiocyanates from Cruciferous Vegetables in Bladder Cancer

**DOI:** 10.3390/molecules25030575

**Published:** 2020-01-29

**Authors:** Tomhiro Mastuo, Yasuyoshi Miyata, Tsutomu Yuno, Yuta Mukae, Asato Otsubo, Kensuke Mitsunari, Kojiro Ohba, Hideki Sakai

**Affiliations:** Department of Urology, Nagasaki University Graduate School of Biomedical Sciences, Nagasaki 852-8501, Japan; tomozo1228@hotmail.com (T.M.); t.yuno@nagasaki-u.ac.jp (T.Y.); ytmk_n2@yahoo.co.jp (Y.M.); a.06131dpsc@gmail.com (A.O.); ken.mitsunari@gmail.com (K.M.); ohba-k@nagasaki-u.ac.jp (K.O.); hsakai@nagasaki-u.ac.jp (H.S.)

**Keywords:** allyl isothiocyanate, benzyl isothiocyanate, sulforaphane, phenethyl isothiocyanate, bladder cancer

## Abstract

Bladder cancer (BC) is a representative of urological cancer with a high recurrence and metastasis potential. Currently, cisplatin-based chemotherapy and immune checkpoint inhibitors are used as standard therapy in patients with advanced/metastatic BC. However, these therapies often show severe adverse events, and prolongation of survival is unsatisfactory. Therefore, a treatment strategy using natural compounds is of great interest. In this review, we focused on the anti-cancer effects of isothiocyanates (ITCs) derived from cruciferous vegetables, which are widely cultivated and consumed in many regions worldwide. Specifically, we discuss the anti-cancer effects of four ITC compounds—allyl isothiocyanate, benzyl isothiocyanate, sulforaphane, and phenethyl isothiocyanate—in BC; the molecular mechanisms underlying their anti-cancer effects; current trends and future direction of ITC-based treatment strategies; and the carcinogenic potential of ITCs. We also discuss the advantages and limitations of each ITC in BC treatment, furthering the consideration of ITCs in treatment strategies and for improving the prognosis of patients with BC.

## 1. Introduction

Bladder cancer (BC) is recognized as a representative of urological cancer (UC), but it has specific pathological characteristics and treatment strategies. BC shows a high recurrence and metastasis potential, even if radical operation is performed [1,2]. Regarding treatment strategies, only two regimens have been approved as effective methods: platinum-based chemotherapy, including gemcitabine and cisplatin combination therapy and MVAC therapy, and immune checkpoint inhibitors, such as pembrolizumab [3,4,5]. Unfortunately, these regimens have shown relatively severe adverse events, including renal dysfunction, neutropenia, and immunogenic abnormalities, and it is currently difficult to predict therapeutic effects [3,4,6]. Although there is an improved prognosis of BC patients undergoing these therapies, prolongation of survival is far from satisfactory, especially in patients with advanced/metastatic disease. Currently, new anti-cancer drugs for UC are under development, some of which are expected to be approved for the treatment of advanced/metastatic BC in the near feature [7,8]. However, there is little information on the safety and adverse events of these new drugs. 

Based on these facts, as well as the artificially produced anti-cancer drugs, treatment strategies employing natural compounds have garnered special interest for various types of malignancies. There is a general consensus that considering treatment strategies using natural product(s) is important for improving the quality of life and prognosis of cancer patients. In fact, there is a report that curcumin, a natural occurring polyphenol derived from turmeric (*Curcuma longa*), had anti-cancer effects via suppression of cancer cell proliferation, invasion, and metastasis in lung cancer cells [9]. The authors also showed that regulation of microRNA expression plays important roles in the curcumin-induced anti-cancer effects [9]. Furthermore, Yiqi Huayu Jiedu decoction, which comprises various Chinese herbs and natural compounds, was reported to increase the anti-cancer effects of standard chemotherapy in patients with stage III gastric cancer after radical gastrectomy and improve the quality of life of patients [10]. In addition to these reports, several reviews and *in vivo* and *in vitro* studies have demonstrated that a variety of natural products possess anti-cancer potential and can avoid the undesirable effects of standard therapies of malignancies [11,12,13,14]. 

Regarding BC, multiple studies showed that a variety of natural products have anti-cancer effects and maintain quality of life, including *Evodia rutaecarpa* or curcumin [15,16]. Several reports also demonstrated the relationship between natural products and prevention of carcinogenesis, clinicopathological features, and anti-cancer effects in BC [17,18]. In previous studies, we showed the anti-cancer effects, clinical usefulness, and pathological mechanisms of green tea polyphenol or royal jelly in urological cancers, including BC [19,20,21,22,23,24,25]. Although green tea and royal jelly are eaten in some parts of Asia and Western countries, this is not the case globally. In contrast, cruciferous vegetables, such as broccoli, kale, cauliflower, bock choy, and horseradish, are widely cultivated in many regions and are commonly eaten worldwide. Isothiocyanates (ITCs) are naturally occurring products of cruciferous vegetables, and researchers have investigated their health benefits and efficacy in the treatment of various diseases, including malignancies. Although some reviews mentioned the anti-cancer effects of ITCs in BC, there is relatively little comprehensive information on the molecular mechanisms of the ITC anti-cancer effects in BC cells [26,27,28]. In addition, comprehensive information on each ITC member, including allyl isothiocyanate (AITC), benzyl isothiocyanate (BITC), sulforaphane (SFN), and phenethyl isothiocyanate (PEITC), is limited. Therefore, in this review, we discuss the anti-cancer effects and efficacy of ITCs in BC cells obtained by *in vivo* and *in vitro* studies. In particular, we focus on the changes in malignant behaviors and cancer-related molecules by ITC members in BC cells. Furthermore, we provide future direction of ITC-based therapy for patients with BC.

## 2. Isothiocyanates in Cruciferous Vegetables

Cruciferous vegetables are classified in the family Brassicaceae/Cruciferae. Several *in vivo*, *in vitro*, and epidemiological studies have shown that cruciferous vegetables inhibit carcinogenesis of BC [27,29,30]. However, in contrast to these reports, a study suggested that cruciferous vegetable intake is not significantly associated with reduced BC risk [31]. Thus, there are controversial results regarding the relationship between cruciferous vegetables and cancer risk in UC. Additionally, there is limited knowledge on the molecular mechanisms of the anti-cancer effects of cruciferous vegetables. 

ITCs are naturally present in cruciferous vegetables and are produced by the hydrolysis of glucosinolates [32]. There is a general agreement that ITCs are beneficial for human health via various mechanisms, such as their anti-microbial activity, prevention of cardiovascular disease, and improvement of fasting glucose levels [33,34]. ITCs are also reported to exhibit anti-carcinogenic activities in various cancer types, including BC [27,33,35,36]. ITCs include the compounds AITC, BITC, SFN, and PEITC (Figure 1), each of which has multiple activities, including anti-cancer effects [37]. In the following sections, we will present the changes in malignant aggressiveness and molecular expression/activity in BC by each ITC member.

### 2.1. Allyl Isothiocyanate

AITC is a volatile and water-insoluble compound derived from various cruciferous vegetables that exhibits multiple functions, such as anti-inflammation, neuroprotection, and anti-bacterial activity [38,39]. In addition, AITC is reported to have anti-cancer effects in various types of malignancies [40,41,42]. However, a study showed that AITC had no significant inhibitory effects on cell proliferation or stimulation of apoptosis in the human breast cancer cell line MDA-MB-231 [43]. Nevertheless, other studies have highlighted the potential advantages of AITC in BC treatment, as the major route of excretion of orally administered AITC is through urine, demonstrating relatively high bioavailability in urine and bladder tissue compared with other organs [44,45]. In this section, we will introduce the anti-cancer effects of AITC in BC and provide future direction for novel treatment strategies using AITC. 

#### 2.1.1. In Vitro Studies

The anti-cancer effects of AITC and its molecular mechanisms have been investigated in various BC cell lines; for instance, AITC was found to lead to morphological changes and inhibit the cell proliferation of the human BC cell lines RT4 and T24 [46]. AITC was also reported to affect cell cycle arrest and apoptosis of RT4 and T24 cells [44]. Moreover, the cytotoxic effects of AITC were confirmed in another BC cell line (UM-UC-3 cells) [47], and the percentages of apoptotic cells increased in an AITC dose-dependent manner in three different BC cell lines (UM-UC-3, UM-UC-6, and T24) [48]. This same study demonstrated that AITC-induced apoptosis is mediated by a mitochondrion-mediated system, including activation of caspase-9, caspase-3, lamin B1, and poly ADP-ribose polymerase (PARP) as well as Bcl-2 phosphorylation at Ser-70 by c-Jun N-terminal kinase (JNK) [48].

These anti-cancer effects, such as anti-proliferation and pro-apoptosis, of AITC are speculated to be independent from TP53—an important regulator of cell death—because RT4 cells possess wild-type TP53, while T24 and UM-UC-3 cells possess mutated TP53 [49]. Moreover, the anti-cancer effects of AITC involved different molecular mechanisms between RT4 and T24 cells. In RT4 cells, AITC treatment increased S100P and Bax levels and decreased Bcl-2 levels; meanwhile, Bax, Bcl-2, and anillin levels increased while S100P levels decreased in T24 cells [46]. The Bax/Bcl-2 pathway is speculated to be a key modulator of AITC in RT4 cells, with anillin and S100P mainly functioning in this system [46]. Thus, these *in vitro* studies demonstrated that the anti-cancer effects of AITC in BC cells are dependent on the pathological and molecular characteristics of cancer cells. 

Molecular mechanisms of the AITC-induced anti-cancer effect in BC cells are shown in Table 1. To our knowledge, there are only two *in vitro* studies on this topic, warranting further research to discuss treatment strategies using ITCs.

#### 2.1.2. *N*-acetylcysteine Conjugate Allyl Isothiocyanate

AITC is mainly excreted in urine as *N*-acetylcysteine conjugate (NAC-AITC). In human and rat BC cells (UM-UC-3 and AY-27 cells, respectively), NAC-AITC inhibits cell proliferation and regulates cell cycle arrest and apoptosis [50]. The anti-cancer effects of NAC-AITC were found associated with downregulation of α-tubulin, β-tubulin/ and vascular endothelial growth factor and activation of caspase-3. Moreover, the authors conclude that the anti-cancer effects of NAC-AITC, including prevention and treatment of cancer are superior to AITC in terms of pharmacokinetic and physical properties. Similar anti-tumor growth activity was found in an orthotopic rat BC model, wherein bladder tumor weight in the NAC-AITC group is significantly lower than that in control (p = 0.0213) [50]. In addition, NAC-AITC suppressed muscle invasion of BC cells (NAC-AITC group = 30%; control = 79%). Similar to the BC cell lines, α- and β-tubulin, vascular endothelial growth factor, and cleaved caspase-3 were found associated with the *in vivo* anti-cancer effects [50].

#### 2.1.3. In Vivo Studies

Dietary administration of a freeze-dried, aqueous extract of broccoli sprouts that included AITC was found to reduce the incidence, multiplicity, and size of BC in an *N*-butyl-*N*-(4-hydroxybutyl) nitrosamine (BBN)-induced rat BC model [51]. However, the detailed molecular mechanisms of this anti-cancer effect were not clearly defined. Another study showed that oral intake of AITC-rich mustard seed powder inhibits tumor growth and muscle invasion in an orthotopic rat BC model via regulation of apoptosis, cell cycle, and angiogenic potential [52]; downregulation of vascular endothelial growth factor and cyclin B1 and upregulation of caspase-3 and cleavage of PARP were found associated with these anti-cancer effects [52]. 

When AITC is stably stored as its glucosinolate precursor (sinigrin) in mustard seed powder (MSP-1), a study revealed that sinigrin itself is not bioactive, whereas hydrated MSP-1 leads to apoptosis and G2/M phase arrest in bladder cancer cell lines *in vitro*. In an orthotopic rat bladder cancer model, oral MSP-1 inhibited bladder cancer growth by 34.5% (P < 0.05) and blocked muscle invasion by 100%. The anti-cancer activity of AITC delivered as MSP-1 appears to be more robust than that of pure AITC. Therefore, MSP-1 may be an attractive delivery vehicle for AITC, as it strongly inhibits bladder cancer development and progression [52].

#### 2.1.4. Combination Therapy of Allyl Isothiocyanate and Conventional Anti-cancer Agents

The cyclooxygenase (COX)-2-plastaglandin (PG) E2-system is an important pathological mechanism of carcinogenesis, tumor growth, and progression in UC [53,54,55]. Therefore, COX-2 inhibitors have been suggested as chemoprotective and therapeutic agents in a variety of cancers [56,57,58]. The synergistic effects of a combination of COX-2 inhibitors and other standard therapy have also been reported [59,60]. Celecoxib, a selective COX-2 inhibitor, is used for various pathological conditions worldwide. Thus, to clarify the anti-cancer effects of a combination of celecoxib and AITC, *in vitro* studies employing AY-27 bladder cancer cells and *in vivo* studies with the F344/AY-27 rat bladder urothelial cell carcinoma model were performed [61]. *In vitro*, AITC first showed no significant impact on COX-2 expression, and PGE2 production was confirmed. However, when the growth inhibitory effects of AITC and celecoxib were analyzed, growth inhibition of AY-27 cells by AITC was not altered by celecoxib addition. The authors thus speculated that the COX-2-mediated anti-tumor growth effects of celecoxib did not reach detectable levels due to excessive dilution of PGE2 in the culture medium. On the other hand, *in vivo* studies employing an animal model with orthotopic BC showed that combination therapy of AITC (1 mg/Kg) and celecoxib (10 mg/Kg) suppresses tumor growth and muscle invasion and that these anti-cancer effects are stronger compared with those of AITC or celecoxib alone. Inhibition of tumor-related angiogenesis regulated by vascular endothelial growth was found to play a crucial role in these anti-cancer effects. 

Another combination therapy employed AITC and cisplatin, a standard anti-cancer drug for patients with BC [62]. *In vitro* studies with lung cancer cells (HOP62) and ovarian cancer cells (2008) showed that the variabilities of both cancer cell lines are significantly inhibited by a combination of AITC and cisplatin, whose inhibitory effects are stronger compared with those of AITC or cisplatin alone. The anti-proliferative effects were confirmed by colony formation assays, and when relationships between cell death and the combination therapy were examined, levels of pro-apoptotic molecules (caspase-3) were found increased and anti-apoptotic molecules (Bcl-2 and survivin) decreased; thus, this combination can suppress tumor growth *in vitro*. Mechanistically, regulation of cell cycle, β-tubulin depletion, and microtubule dysfunction are associated with the anti-cancer effects of AITC and cisplatin. Finally, the combination index of ATIC and cisplatin in lung cancer cells indicates a synergistic interaction. Indeed, *in vivo* studies with A549-derived lung cancer xenograft tumor models showed decreases in tumor volumes after combination therapy (AITC = 50 mg/Kg and cisplatin = 6 mg/kg), whereas tumor volumes increased after AITC (50 mg/Kg) or cisplatin (6 mg/kg) monotherapy. Overall, the anti-cancer effect parameters (i.e., maximum tumor growth inhibition, tumor doubling time, and frequency of partial response and complete response) are remarkably better after combination therapy than after either monotherapy. Furthermore, AITC + cisplatin therapy exhibits no toxicity, including maximum weight loss of pretreatment bodyweight. 

#### 2.1.5. Clinical Trials and Future Direction of Allyl Isothiocyanate-Based Therapy

Recently, an *in vitro* study employing the macrophage cell line RAW 264.7 and human BC cell line HT1376 was conducted to clarify the anti-inflammatory activity and anti-cancer effect of AITC nanoparticles [63]. The results showed that AITC nanoparticles inhibit cancer cell proliferation and migration; however, these anti-cancer effects are dependent on AITC concentration; inhibition of cancer cell proliferation and migration is achieved at 70 mg L^−1^ and 8.75 mg L^−1^ of AITC nanoparticles, respectively. AITC nanoparticles were also found to suppress production of lipopolysaccharide-induced tumor necrosis factor (TNF)-α, interleukin (IL)-6, nitric oxide (NO), and inducible NO synthase in macrophage cells, and their anti-inflammatory effects are stronger than those of AITC or nanoparticles alone. The authors thus suggested that AITC nanoparticles can be a valuable treatment strategy for BC via their regulation of inflammation, immunity, and oxidative stress. 

Other novel strategies employing AITC are currently under development. For example, the anti-cancer effects of AITC-conjugated silicon quantum dots were examined in human umbilical vein endothelial cells (HUVECs) and human hepatocellular carcinoma cells (HepG2) [64]. Interestingly, high doses of AITC (40–320 μM) were found to significantly inhibit HepG2 cell viability, whereas low doses (5 μM) significantly stimulated cancer cell viability. Similar trends were confirmed for cancer cell migration (inhibition at 20 μM AITC and stimulation at 2.5 μM) and angiogenesis (HUVEC tube formation ability is suppressed at > 5 μM but stimulated at even lower doses of 1.25 and 2.5 μM AITC). Thus, there is a possibility that the anti-cancer effects of AITC are dependent on its concentration and that low concentrations of AITC may have detrimental effects via increased cancer cell proliferation, migration, and angiogenesis in hepatocellular carcinoma. The authors further showed that AITC-conjugated silicon quantum dots overcame the limitation of AITC in the same analysis. Therefore, AITC-conjugated silicon quantum dots are suggested as a useful drug delivery system for AITC in cancer patients. Although there are no data in BC, AITC is also predicted to have biphasic effects of anti-cancer effects and angiogenesis. Therefore, we suggest additional *in vivo* and *in vitro* studies of the silicon quantum dot system in BC to elucidate new treatment strategies for patients. 

### 2.2. Benzyl Isothiocyanate

Similar to other ITC members, BITC has immunomodulatory, anti-microbial, and anti-oxidative activities under various pathological conditions [34,65,66] Several studies have also shown that BITC possesses anti-cancer and chemopreventive effects in various types of malignancies [67,68,69]. However, there is limited information on its anti-cancer effects and molecular mechanisms in UC. 

#### 2.2.1. In Vitro Studies

Similar to AITC, BITC has shown anti-proliferative and pro-apoptotic activity in BC cells [70,71]; however, the pro-apoptotic activity of BITC is stronger compared with that of other ITC members, including AITC and SFN. Moreover, caspase-9 is the main regulator of BITC-induced apoptosis in UM-UC-3 cells, although all ITC members exhibit pro-apoptotic activities via activation of caspase-3, 8, and 9 [70,72]. Additionally, mitochondrial activities are targets of BITC, and BITC-induced changes are regulated by various members of the Bcl-2 family, including Bcl-2, Bax, Bak, and Bcl-xl [70]. 

As previously mentioned, ITCs are primarily disposed and concentrated in the urine as NAC conjugates. UC originates from urothelial cells and is constantly exposed to urine in the urinary tract. Therefore, studies have focused on the anti-cancer effects of NAC-conjugated BITC in BC cells [71] and found that it suppresses BC cell growth through anti-proliferative and pro-apoptotic activities. Activation of caspase-3, 8, and 9; cell cycle arrest in phases S and G2/M; and regulation of Cdc25C were associated with the anti-proliferative function of NAC-conjugated BITC. The authors confirmed, however, that longer treatment durations or higher doses of NAC-conjugated BITC are necessary to exert similar effects as those of BITC. 

miRNAs are major modulators of carcinogenesis, malignant aggressiveness, and outcome in UC [73,74] and several miRNAs are closely associated with cisplatin sensitivity of BC cells [75]. miR-99a-5p, a tumor suppressor, exhibits anti-proliferative and pro-apoptotic activities in UC [76,77,78]. One study demonstrated that BITC treatment upregulates miR-99a-5p expression in the BC cell lines 5637 and T24 [76], which leads to decreased mRNA and protein levels of IGF-1R, FGF-R3, and mTOR in both BC cell lines. The authors also demonstrated a molecular mechanism associated with regulation of BC cell survival and apoptosis by BITC. Taken together, these findings indicate that BITC exhibits anti-cancer effects via regulation of cell survival in UC. Another study elucidated the anti-cancer effects of BITC-induced miR-99a expression in BC cells [79] and reported that BITC enhances miR-99a expression in 5637 and T24 BC cells, which is associated with ERK activation and nuclear transcriptional activation of c-Jun/(activator protein) AP-1. Thus, the authors suggested that BITC stimulates miR-99a expression via regulation of the ERK/AP-1 pathway in BC and demonstrated the anti-cancer effects of miR-99a in UC. Nevertheless, there is a general consensus that the anti-carcinogenic and anti-cancer effects of miR-99a represent a complex mechanism in BC cells [80,81,82]. Therefore, this information is useful for understanding the biological function of BITC in BC. The molecular mechanisms of the anti-cancer effects of BITC are shown in Table 2. 

#### 2.2.2. In Vivo Studies

In a rat model of BBN-induced BC, oral intake of BITC suppressed the incidence of neoplastic pathological changes, such as dysplasia, papilloma, and carcinoma, and multiplicities in a dose-dependent manner (10, 100, or 1,000 ppm BITC) [83]. Notably, epithelial hyperplasia of the bladder was found in rats treated with 100 or 1,000 ppm BITC without BBN [83]. The same researchers also demonstrated the carcinogenic potential of BITC in this BC animal model [84]. Therefore, the toxicity and risk of BITC in BC treatment should be considered; this is further detailed later in the text (see Section 2.1).

#### 2.2.3. Combination Therapy of Benzyl Isothiocyanate and Cisplatin

As mentioned in Section 2.4, combination therapy of ITCs and cisplatin is expected to have better anti-cancer effects than those of ITCs or cisplatin alone. Indeed, several reports showed that BITC enhances the anti-cancer effects of cisplatin in lung cancer cells (NCI-H596), head and neck squamous cell carcinoma cells (HN12, HN8, and HN30), and leukemia cells (HL-60) [85,86,87]; there are no similar studies in BC cells, however. 

### 2.3. Sulforaphane

SFN can be found in cruciferous vegetables, such as broccoli, cauliflower, brussel sprouts, cabbage, kale, and kohlrabi [88]. SFN is reported to regulate cancer cell survival via inhibition of cell proliferation and stimulation of apoptosis in a variety of cancers [89,90]. Among the ITC members, SFN has been the most widely investigated regarding its pathological roles and molecular mechanisms both *in vivo* and *in vitro*. 

#### 2.3.1. In Vitro Studies: Cell Cycle-, Caspase- and Bcl-2-Related Molecules

Regarding the relationships between SFN and cell survival, including cell proliferation, cell cycle, and death, various mechanisms have been suggested. For example, SFN was reported to induce growth arrest and apoptosis in a BC cell line (5637 cells) [91]. Moreover, induction and stimulation of cyclin B1 and Cdk1 were found associated with the anti-proliferative effects of SFN, whereas activation of caspase-3, 8, and 9 and PARP corresponded to its pro-apoptotic effects; these SFN-induced anti-cancer effects are speculated to be regulated via reactive oxygen species (ROS)-dependent mechanisms [91]. Another study showed that SFN treatment suppresses cell viability in a dose-dependent manner and induces apoptosis in T24 human BC cells via regulation of caspase-3, caspase-9, and PARP [92]. Moreover, the SFN-induced apoptosis of BC cells is mediated by dysregulation of mitochondria function, cytochrome *c* release, and Bcl-2-related pathways [92]. 

Other studies focused on the relationships between cell cycle-related molecules and SFN. For instance, after 10–40 μM SFN treatment for 24 or 48 h, T24 cell viability is significantly suppressed with IC50 values of 26.9 ± 1.12 μM (24 h) or 15.9 ± 0.76 μM (48 h) [93]. Conversely, 20 μM SFN treatment for 24 or 48 h resulted in apoptotic features, such as cell shrinkage, condensed chromatin, and apoptotic bodies, in the same BC cells; increased numbers of apoptotic cells were confirmed by flow cytometry [93]. SFN is also associated with blocking cell cycle progression at G0/G1 phase. In addition to its pro-apoptotic activities, upregulation of the cyclin-dependent kinase inhibitor p27 plays crucial roles in the 20 μM SFN-induced anti-cancer effects in BC cells, whereas p16 or cyclin D1 expression does not [93]. Thus, regulation of cell cycle-related molecules and mitochondrial function, caspases, and the Bcl-2 protein family represent the molecular mechanisms of SFN-induced anti-proliferative and pro-apoptotic activities in BC cells.

#### 2.3.2. In Vitro Studies: Oxidative Stress, Endoplasmic Reticulum Stress, and Growth Factors

As mentioned above, many investigators believe that the anti-cancer effects of SFN in BC are mainly associated with caspase- and mitochondria-related pathways. Nevertheless, there are other cancer-related factors involved. For instance, SFN can inhibit DNA damage induced by chemical carcinogens in BC T24 cells [94]. Moreover, SFN-induced oxidative stress through ROS has been suggested as a key modulator [91,92]. Nuclear factor erythroid 2-related factor-2 (Nrf2) regulation and endoplasmic reticulum (ER) stress are also associated with SFN and carcinogenesis, pathological behavior, and cell survival in UC [28,92]. Notably, these Nrf2 and ER signaling pathways are important factors in the response to oxidative stress and anti-oxidative activities [28,92,95]. A study showed that enhanced insulin-like growth-factor-binding protein-3 (IGFBP-3) and suppressed nuclear factor-kappa B (NF-κB) expression by SFN are associated with the anti-proliferative effect of SFN in the BC cell line BIU87. Interestingly, the authors also found that SFN stimulates apoptosis and cell cycle arrest at the G2/M phase, resulting from IGFBP-3 and NF-κB regulation [96]. As IGFBP-3 and NF-κB are known to possess pro-apoptotic and anti-apoptotic functions, respectively, in various malignancies [97,98], this stimulation of apoptosis by SPN via increased IGFBP-3 and decreased NF-κB levels are in agreement with established findings. Another report on the relationship between SFN-induced anti-cancer effects and growth factors demonstrated that 20 μM SFN leads to a 2.6-, 3.0-, or 3.1-fold increase in the G_2_/M phase compared with that of controls in three BC cell lines (RT4, J82, and UM-UC-3, respectively) [99]. In addition, SFN induces apoptosis in RT4 and UM-UC-3 cells. Thus, these findings indicate that upregulation of caspase-3/7 and PARP activity and downregulation of survivin, EGFR, and HER2/neu are the underlying molecular mechanisms. 

TNF-related apoptosis-inducing ligand (TRAIL) is recognized as an initiator of apoptosis. Its dysregulation has been identified in various malignant cells, including BC [100,101]. As a result, resistance to TRAIL is associated with high malignant potential and worse prognosis for patients with BC [102]. SFN treatment however, has been reported to reverse the pro-apoptotic activity of TRAIL in TRAIL-resistant BC cells [103]; the SFN-induced mechanisms were found associated with apoptosis-related molecules (e.g., caspases, mitochondrial membrane potential, Bid, and death receptor 5) and oxidative stress-related factors (e.g., ROS and Nrf2). 

The anti-cancer effects of SFN under hypoxic conditions in BC cell lines have also been reported [88]; in RT112 cells, 20 μM SFN inhibited cancer cell proliferation by 26.1 ± 4.1% and 39.7 ± 5.2% under normoxia and hypoxia, respectively (P < 0.05), with similar results observed for RT4 cells (normoxia, 29.7 ± 4.6%; hypoxia, 48.3 ± 5.2%). Tumor tissues, especially those within the center, are generally under hypoxic conditions due to the oxygen consumption of the tumor to support its growth. Thus, these findings indicate that SFN can suppress cell proliferation under hypoxic conditions in BC with rapid tumor growth compared with that under normoxia and relatively slow growth. Interestingly, the same study also showed that SFN suppresses glycolytic metabolism under hypoxia by decreasing the nuclear translocation of hypoxia-inducible factor-1α, thereby reducing its protein levels [88]. Suppression of glycolytic metabolism in cancer cells is important for inhibiting tumor growth and progression as high glycolytic metabolism leads to increased cancer cell proliferation. Overall, the findings demonstrate that SFN plays several roles in suppressing malignant aggressiveness, such as by decreasing cancer cell proliferation, in BC cells. 

#### 2.3.3. In Vitro Studies: Inflammation, Epithelial-to-Mesenchymal Transition, Epigenesis, and Others

In addition to reducing BC cell survival, SFN inhibits malignant aggressiveness by suppressing inflammation, cancer cell invasion, and metastasis. Several studies have shown that SFN downregulates COX-2 expression in BC cells via regulation of p38 mitogen-activated protein kinase (MAPK) and NF-κB [104,105,106]. Moreover, p38 MARK is positively associated with glutathione transferase and thioredoxin reductase-1—both antioxidant enzymes—following SFN treatment [105]. Furthermore, SFN can inhibit epithelial-to-mesenchymal transition (EMT)—an important mediator of cancer cell invasion and metastasis—via regulation of COX-2/matrix metalloproteinase (MMP)-2, -9/ZEB1, Snail, and miR-200c/ZEB1 in BC cells [106]. 

SFN was found to inhibit histone status in BC cells, which is associated with reduced levels of histone H1 phosphorylation via modification of histone acetyltransferase and histone deacetylase activity [26]. Changes in histone H1 status were previously reported to be associated with carcinogenesis and prognosis of BC [107]. Based on these findings, SFN is speculated to inhibit carcinogenesis and progression of BC via epigenetic modification [26].

Recently, the physiological and pathological roles of gut microbiota have garnered great interest. Research has shown how they affect systematic metabolism, inflammation, and the immune system, contributing to carcinogenesis, malignant potential, and cancer progression, of which similar findings have been reported in UC [108,109,110]. Interestingly, SFN was found to normalize gut microbiota dysbiosis by increasing the abundance of *Bacteroides fragilis* and *Clostridium* cluster I in a BBN-induced BC animal model [111], suppressing BBN-induced histological changes, including sub-mucosal capillary growth. While the detailed mechanisms of the anti-carcinogenic function of SFN in this model is not fully clear, normalization of intestinal flora has been shown to repair intestinal barrier dysfunction and injured mucosal epithelium via regulation of tight junction proteins, including ZO-1, claudin-1, occludin, and mucin-2 [111]. Moreover, SFN plays crucial roles in the inflammatory status of this model, as it decreases pro-inflammatory factors such as IL-6 and secretory immunoglobin A, which are increased by carcinogenesis [111]. The authors conclude that these gut microbiota-related beneficial effects of SFN led to its anti-carcinogenic effects in BC via complex mechanisms that involve inflammation and the immune system [111]. A summary of the molecular mechanisms of the anti-cancer effects of SFN is shown in Table 3.

#### 2.3.4. In Vivo Studies

*In vivo* studies with the chemical-induced BC animal model have shown that SFN inhibits carcinogenesis, tumor growth, and progression via a complex mechanism that includes prevention of DNA damage [94]. In a murine UM-UC-3 xenograft model, tumor growth rates and tumor weights in the SFN group were found lower than in the control group (not significant and *p* < 0.05, respectively) [51]. Furthermore, this model showed decreases in tumor volumes in SFN-treated mice (12 mg/kg bodyweight for 5 weeks) with an inhibitory rate of 63% via increased caspase-3 and cytochrome *c* expression and decreased survivin expression [113]. In addition to the apoptosis-related pathways, several other molecules have been suggested to be associated with the anti-cancer effects of SFN, based on *in vivo* studies. Thus, further *in vivo* studies are essential for understanding the efficacy and limitations of an SFN-based treatment strategy against BC.

#### 2.3.5. Combination Therapy of Sulforaphane and Other Therapeutic Agents

Although several clinical trials on the anti-cancer effects of SFN and broccoli sprout extracts have been performed, the results were unsatisfactory [114,115]; for example, no or minimum effects are detected on serum and tissue biomarkers of patients with prostate and breast cancer. Although a similar clinical trial has not been performed for patients with BC, the clinical effects of SFN monotherapy are also predicted to be unsatisfactory. Therefore, the efficacy of a combination therapy of SFN and other therapeutic agents was investigated in BC cells.

A combination therapy of acetazolamide (AZ; a carbonic anhydrase inhibitor) and SFN showed suppressed proliferative and clonogenic effects and stimulated apoptotic activity via caspase-3 and PARP activation [116]. In addition, the PI3K/Akt signaling pathway was found to play an important role in the anti-cancer effects of this combination therapy. The authors thus conclude that AZ + SFN is a potential therapeutic strategy for BC. Another study examined the effects of two ITCs (AITC + SFN) on the lung cancer cell line A549 [117]; their anti-carcinogenic effects showed higher inhibitory effects on tumor growth and cancer cell migration and greater stimulation of apoptosis compared with that of ATIC or SFN alone. Moreover, oxidative stress, including ROS, is associated with these activities. Although this study was not performed on BC cells, we believe that a combination of different ITCs may be effective for the prevention and treatment of BC. Indeed, in the BC cell line UM-UC-3, pro-apoptotic activity of BITC or PEITC alone is stronger than that of AITC or SFN alone [72]. Moreover, ≥20 μM SFN significantly suppresses cell proliferation in the BC cell line BIU87, whereas 10 μM SFN had no significant effect [96,112]. Therefore, more detailed studies on the combination of various ITC types, dosages, and durations are necessary to identify the most efficacious combination therapy of the ITC members.

Nevertheless, there are potential limitations of SFN-based therapies. Novel immunotherapy strategies, such as immune checkpoint inhibitors, have been recently established as standard therapy for patients with advanced/metastatic BC [118,119]. While we speculate that a combination of immunotherapy and SFN may be useful for the prevention and treatment of BC, a combination of SFN with T cell-mediated cancer immunotherapies is not recommended because SFN can function both as an anti- and pro-carcinogenic factor due to its effects on tumor and immune cells [120]. 

### 2.4. Phenethyl Isothiocyanate

Similar to other ITC members, PEITC can suppress carcinogenesis and malignant aggressiveness in various types of malignancies [121]. Suppression of various cancer-promoting characteristics, such as cancer cell proliferation, invasion, and angiogenesis, via regulation of the Bcl-2 protein family, caspases, and matrix metalloproteinases are reported as the potential molecular mechanisms underlying the tumor-suppressive activities of PEITC [121,122,123].

#### 2.4.1. In Vitro Studies

PEITC was shown to have anti-cancer regulatory effects on cancer cell survival and apoptosis in BC cells [124]; PEITC inhibits cell viability in a dose-dependent manner and enhances apoptotic potential, as measured by caspases activities in T24 cells [124]. However, as shown in Table 4, the detailed molecular mechanisms underlying these anti-cancer effects in BC cells are not fully understood. It was reported that PEITC inhibits cell proliferation and stimulates apoptosis in the human adriamycin (ADM)-resistant bladder carcinoma cell line T24/ADM [123]. Interestingly, this study showed that PEITC increases intracellular drug accumulation potential and DNA topoisomerase II expression, and decreases multidrug resistance-related factors, such as multidrug resistance gene (MDR1), multidrug resistance-associated protein (MRP1), and glutathione S-transferase π [123]. In general, such changes by PEITC lead to increased chemosensitivity. Additionally, the authors clarified the detailed molecular mechanism underlying multidrug resistance reversal potential, which includes downregulation of NF-κB, survivin, Twist, and Akt and upregulation of PTEN and JNK by PEITC. As chemotherapeutic regimens including ADM are the standard therapy for patients with advanced UC [125,126], these findings highlight PEITC as a potential therapeutic agent for BC treatment, especially in patients with drug-resistant BC [123]. While we agree with their conclusion, clinical trials testing this hypothesis have yet to be performed. 

Although BITC has been suggested to suppress cell growth by upregulating miR-99a expression via regulation of the c-Jun/AP-1 pathway in BC [79,80], this pathway plays no significant role in the anti-proliferative effects of PEITC in BC cells [127]. *In vitro* molecular mechanisms of the anti-cancer effects of PEITC are shown in Table 4.

#### 2.4.2. In Vivo Studies

PEITC is suggested to play crucial roles in preventing the initiation step of carcinogenesis and inhibiting tumor progression in a variety of malignancies [121]. However, in a chemically BBN-induced BC animal model using male human c-Ha-ras proto-oncogene transgenic rats, microscopic BC is observed in the BBN alone (16 weeks) and BBN (8 weeks) → PEITC (8 weeks) groups; but not in the PEITC (8 weeks) → BBN (8 weeks) group [128]. This finding indicates that PEITC can inhibit the carcinogenic process after initiation. However, a conclusion cannot be drawn due to the limited information on *in vivo* SFN activities in BC. 

#### 2.4.3. Combination Therapy of Phenethyl Isothiocyanate and Other Therapeutic Agents

We previously introduced a novel treatment strategy that combines AITC and cisplatin for lung cancer cells (Section 2.1.4). Similarly, the efficacy of a combination therapy of PEITC and cisplatin was demonstrated in several studies. For instance, cervical cancer cells (HeLa) treated for 24 h with 5 μM PEITC and 10 μM cisplatin show typical features of apoptosis, such as cell shrinkage, membrane blebbing, and cell detachment, with a 4-fold increase in caspase-3 activity; these significant changes are not observed for either treatment alone [122]. The same study also showed that PEITC increases the pro-apoptotic activity of cisplatin in C33A cervical cancer and MCF-7 breast cancer cells; interestingly, this pro-apoptotic activity is not detected in normal human mammary epithelial MCF-10A cells [122]. Another study on non-small cell lung cancer cells (A549) showed that the percentage cell survival after treatment with AITC (15 μM) or cisplatin (5 μM) alone is 79.2 ± 3.8% and 55.9 ± 3.4%, respectively, whereas cell survival in the combination group is 46.2 ± 2.7% [129]. Notably, when PITC and cisplatin are co-encapsulated in liposomal nanoparticles, A549 cell survival further decreased to 33.3 ± 2.9% [129]. Similar results were obtained in another non-small cell lung cancer cells (H596), where the percentage cell survival after treatment with liposomal-PEITC-cisplatin or free PITC + cisplatin is 55.0 ± 9.5% and 28.6 ± 6.3%, respectively (p < 0.001) [129]. Moreover, the liposomal nanoparticles containing both PEITC and cisplatin have the advantage of increased circulation time in the bloodstream and accumulation in tumors [130]. Therefore, co-encapsulated PITC and cisplatin in liposomal nanoparticles may be a potential therapeutic strategy for advanced/metastatic UC. 

## 3. Carcinogenic Potential of Isothiocyanates

There is general consensus that all ITC members possess anti-cancer effects in BC cells. However, several studies have also suggested the carcinogenic potential of ITCs in BC. In this section, we will discuss the relationships between BITC, SFN, and PEITC and carcinogenic changes in BC. To our knowledge, AITC has not been shown to promote tumorigenesis and carcinoma in BC; nevertheless, we cannot conclude that AITC has no carcinogenic potential as there is limited information on the biological and pathological effects of AITC in BC.

### 3.1. Carcinogenic Potential of Benzyl Isothiocyanate

In a two-stage carcinogenesis model, rats treated with BITC and with BBN initiation show neoplastic lesions, including papillary or nodular-hyperplasia (100%), papilloma (38%), and carcinoma (100%); these frequencies are higher than in rats under a basal diet (57%, 5%, and 24%, respectively) [131]. The frequencies of papilloma and carcinoma are also lower than those in rats with initiation + BITC (papilloma = 17% and carcinoma = 0%) and rats without initiation. Therefore, BITC may enhance the carcinogenic process in rats with initiation alone. However, in a BBN-induced BC rat model, oral administration of 10, 100, or 1,000 ppm BBN suppresses carcinogenic pathological changes [83]; moreover, epithelial hyperplasia of the urinary bladder is detected in rats treated with 100 or 1,000 ppm BITC, even without BBN [83]. Furthermore, the same research group showed that these neoplastic changes increase in rats with initiation treatment of 500 ppm BBN and subsequent low dose (25 ppm) BBN exposure, and their frequencies are further increased by additional treatment with 100 and 1000 ppm BITC in a dose-dependent manner [84]. In a similar experiment without initiation treatment, dysplasia, papilloma, and carcinoma were rare, although almost all rats had hyperplasia, except for the control and 100 ppm BITC groups [84]. Thus, BITC may stimulate carcinogenesis in a high-risk population of BBN-induced BC cases [83,84,132]. 

### 3.2. Carcinogenic Potential of Sulforaphane

As shown in a previous study, ≥20 μM SFN decreases cell viability and migration of T24 BC cells [112]. However, the study also showed that a low concentration of SFN promotes BC cell proliferation and migration [112], where 1–5 μM SFN or 2.5 and 3.75 SFN increase cell growth to approximately 120–130% and cell migration to 128 and 133% compared with those of control [112]. Thus, a biphasic effect of SFN on cell growth and migration of BC cells was suggested. Mechanistically, activation of autophagy by SFN is speculated to be associated with upregulated cell migration in an *in vivo* study using the autophagy inhibitor 3-methyladenine in T24 cells. The authors also found an enhanced protective effect in conjunction with selenium against free radical-induced cell death. Although this mechanism was confirmed in human hepatocyte cells (HHL-5) and breast cancer cells (MCF-7) rather than in BC cells, the benefits, and risks of SFN have been shown to be dependent on its doses and interactions with the microenvironment, including autophagy and selenium.

### 3.3. Carcinogenic Potential of Phenethyl Isothiocyanate

In an animal model of dimethylbenzanthracene-induced mammary carcinogenesis, continuous oral administration of 1200 ppm PEITC induces hyperplasia in the urinary bladder [133]. However, another study showed a high frequency of carcinoma (11 of 12 rats; 91.7%) with oral administration of 0.1% PEITC in rats for 48 weeks [134]; the authors thus conclude that PEITC has carcinogenic activities in the rat urinary bladder. By contrast, in a two-stage carcinogenesis model, rats treated with PEITC with BBN initiation exhibit papillary or nodular-hyperplasia (100%), papilloma (24%), and carcinoma (100%). Meanwhile, frequencies of papilloma (17%) and carcinoma (33%) are lower in rats without initiation than in those with initiation [131]. In studies showing the carcinogenic potential of PEITC according to initiation, specifically in a rat medium-term multi-organ carcinogenesis model, oral treatment with 0.1% PEITC after the initiation period leads to incidences of papillary or nodular-hyperplasia and tumors [135]. However, the authors showed that PEITC provided during the initiation period is not associated with carcinogenic activity [135]. These findings suggest that PEITC may stimulate carcinogenesis of UC during the post-initiation period. Another study, however, demonstrated that PEITC increases the incidences of papillary or nodular hyperplasia, dysplasia, and carcinoma in a dose-dependent manner; thus, > 0.01% PEITC enhances rat urinary bladder carcinogenesis and > 0.05% PEITC has tumorigenic potential. [136]. Collectively, these findings indicate that carcinogenic potential of PEITC administration may be modulated by complex mechanisms that involve timing and dosage. 

When 0.1% BITC or 0.1% PEITC is administered in the diet to 6-week-old F344 rats for 1, 2, 3, and 7 days, a significant reduction of urinary pH levels compared to the normal control is detected, starting at day 1 [132]. Similarly, a reduction is detected in the urinary concentration of Na and Cl, whereas K is reduced. The same study also showed that thickness of the urinary bladder urothelium is significantly increased by administration of both BITC and PEITC and that inflammation, vacuolation, erosion, and apoptosis/single cell necrosis occur in the urinary bladder lesion; these morphological changes are not observed in normal control rats [132]. Furthermore, the cell proliferation potential, evaluated by the BrdU labeling index in male rats treated with BITC and female rats with BITC + PEITC, is significantly higher than that of control rats [132]. By contrast, when 0.1% BITC or 0.1% PEITC is administered for 14 days, histopathological simple hyperplasia and papillary/nodular hyperplasia are detected in 100% and 86% and 100% and 60% of the cases, respectively [132]. The authors thus suggest that continuous proliferation of bladder epithelial cells by BITC and PEITC plays important roles in pathological changes, including inflammation and the early stage of carcinogenesis [132]. Taken together, these findings are extremely important for the consideration of ITC treatment strategies, especially BITC and PEITC, for UC. However, we should note the difference in administered levels of ITCs in these studies. Although the mean daily consumption of BITC and PEITC in rats was approximately 80 mg/kg/day [132], these levels do not reflect human physiological conditions, i.e., 30 g of fresh watercress = 7.6 mg of PEITC per person and 0.8 mg from fresh (0.5 mg) and cooked (0.3 mg) Swede-turnips = 0.28 mg/person/day of PEITC [137,138]. 

## 4. Further Considerations

As previously mentioned, ITCs exhibit their anti- and pro-carcinogenic activities via complex mechanisms. In this review, we mainly introduced the findings of pre-clinical *in vivo* and *in vitro* studies for easier understanding across the field. However, we would be remiss if we do not mention that other cancer-related factors and signaling molecules affect the biological activities and anti-cancer effects of ITCs in malignancies. These include direct/indirect interactions with Nrf2 and NF-κB, the Nrf2-Kelch-like ECH-associated protein (Keap) 1-antioxidant response element (ARE) signaling pathway, and antioxidant enzymes, such as NAD(P)H quinone reductase (NQO1) and glutathione S transferases (GSTs), through the Nrf2-Keap1-ARE signaling pathway are closely associated with ITC-induced bioactivity [139,140,141]. We would like to emphasize that further basic research is essential for uncovering the utility and limitations of ITCs in cancer treatment, including BC.

Another important issue to consider is the carcinogenic risk factors of BC, which are affected by a variety of harmful chemical compounds (e.g., cigarette smoke) or physiologically active substances (e.g., sex hormones) [142,143,144]. With regards to cigarette smoke, cytochrome P450 and phase II detoxification enzymes, such as DOQ1, GSTs, and glucuronosyltransferase inhibit the formation of carcinogenic compounds from tobacco-specific carcinogens, and PEITC modulates such cancer preventive activities [145]. This finding supports the hypothesis that PEITC may suppress the tobacco-related cancer risk in smokers. Indeed, a clinical trial showed that metabolic activation of a tobacco-specific lung carcinogen is significantly suppressed by PEIT treatment [146]. We believe that the cancer risk of BC in smokers may be suppressed by ITCs though similar anti-carcinogenic mechanisms, and thus there is value in performing such clinical trials for BC. Furthermore, the frequency of BC is known to be remarkably higher in men than in women, which is perhaps due to the testosterone-androgen receptor pathways [147]. Interestingly, PEITC was reported to suppress testosterone-induced cancer cell proliferation by downregulating the testosterone-androgen receptor pathway in prostate cancer [148]. Meanwhile, other research has shown that estrogen-mediated pathways are associated with malignant potential and tumor growth of BC [143,144]. In addition, SFN was found to regulate tumor growth of breast cancer cells by modulating estrogen activities [149]. Unfortunately, there is little information on the influence of ITC-mediated sex hormone activity on the malignant potential of BC. Nevertheless, there is a possibility that ITCs affect carcinogenesis and malignant aggressiveness by regulating sex hormones in BC. This highlights the need for designing studies to identify the biological roles of ITCs according to patient background and environment, including occupation, diet, and health habits.

## 5. Conclusions

In this review, we discussed the anti-cancer effects of ITCs in BC. The research suggests that all ITC members can suppress carcinogenesis, tumor development, and progression *in vivo* and *in vitro*. Furthermore, regulation of cell proliferation, cell cycle, and apoptosis play crucial roles in the ITC-induced anti-cancer effects, and such phenomena are mainly regulated by complex mechanisms involving caspases, Bcl-2 family proteins, and mitochondrial activities. While changes in cancer-related molecules by ITCs may correspond to anti-cancer mechanisms in BC cells, some ITCs may have neoplastic and carcinogenic potential in BC. To clarify this issue, more detailed studies at the molecular level are essential. While there is a possibility that ITC-based treatment strategies can improve prognosis in patients with BC, further clinical trials with well-designed protocols are required to establish the optimal doses and types of ITCs for application in BC treatment [112]. In addition, it would be fruitful to investigate the anti-cancer effects and clinical utility of combination therapies of ITCs and new therapeutic strategies, including immunotherapy and gene therapy, for patients with BC. 

## Figures and Tables

**Figure 1 molecules-25-00575-f001:**
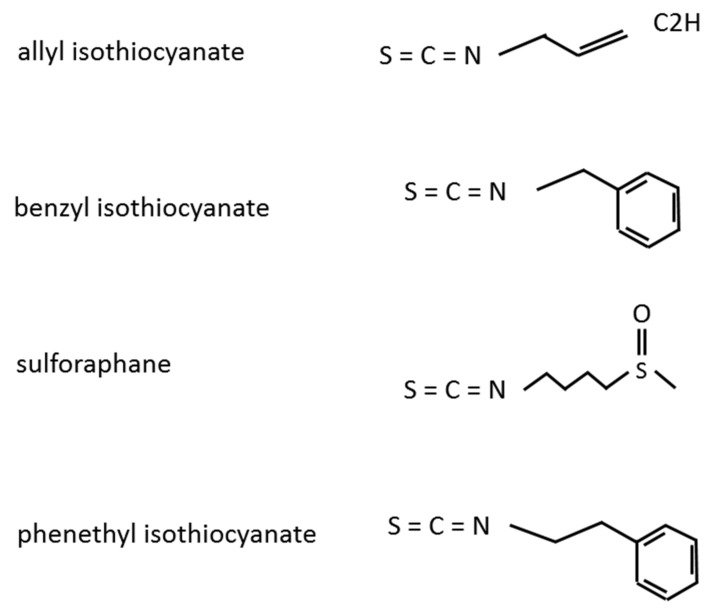
Structures of the isothiocyanate members.

**Table 1 molecules-25-00575-t001:** *In vitro* molecular mechanism of the anti-cancer effects of allyl isothiocyanate.

Anti-Cancer Effect	Underlying Molecular Mechanisms	Reference
Cell growth ↓	Increased S100P and Bax expression and decreased Bcl-2 expression in RT4 cells	Sávio et al., 2015 [46]
Cell growth ↓	Increased Bcl-2, Bax, and anillin expression and decreased S100P expression in T24 cells	Sávio et al., 2015 [46]
Apoptosis ↑	Regulation of mitochondrion-mediated mechanisms and Bcl-2 phosphorylation	Geng et al., 2011 [48]

Bcl, B-cell lymphoma-2; Bax, Bcl-2-associated X protein.

**Table 2 molecules-25-00575-t002:** *In vitro* molecular mechanisms of the anti-cancer effects of benzyl isothiocyanate.

Anti-Cancer Effect	Underlying Molecular Mechanisms	Reference
Cell growth ↓	Suppression of IGF1R, FGFR3, and mTOR activation by miR-99a-5p upregulation	Liu et al., 2019 [76]
Apoptosis ↑	Via caspase-9, a major regulator, and Bcl-2, Bax, Bak, and Bcl-xl	Tang & Zhang, 2005 [70]
Apoptosis ↑	Stimulation of caspase-3, 8, and 9 and cell cycle arrest in the same phases by Cdc25C	Tang et al., 2006 [71]

IGF1R, insulin-like growth factor 1 receptor; FGFR, fibroblast growth factor receptor; mTOR, mechanistic target of rapamycin; Bcl, B-cell lymphoma-2; Bax, Bcl-2-associated X protein; Bak, BCL2-antagonist/killer.

**Table 3 molecules-25-00575-t003:** *In vitro* molecular mechanisms of the anti-cancer effects of sulforaphane.

Anti-Cancer Effect	Underlying Molecular Mechanisms	Reference
Cell growth ↓	Increased IGFBP-3 expression and decreased NF-κB expression	Dang et al., 2014 [96]
Cell growth ↓	Increased cyclin B1 and Cdk1 phosphorylation and their complex effects	Park et al., 2014 [91]
Cell growth ↓	Suppression of HIF-1α-mediated glycolytic metabolism under hypoxic conditions	Xia et al., 2019 [88]
Apoptosis ↑	Increased expression of the cyclin-dependent kinase inhibitor p27	Shan et al., 2006 [93]
Apoptosis ↑	Increased caspase-3/7 and PARP expression and decreased survivin, EGFR, and HER2/neu expression	Abboui et al., 2012 [99]
Apoptosis ↑	Increased IGFBP-3 expression and decreased NF-κB expression	Dang et al., 2014 [96]
Apoptosis ↑	Activation of ROS-mediated caspase-3/9 and PARP, ER stress, and Nrf2	Jo et al., 2014 [92]
Apoptosis ↑	Activation of caspase-3, 8, and 9 and PARP via ROS-dependent pathways	Park et al., 2014 [91]
Apoptosis ↑	Reversal of TRAIL activity via regulation of caspases, MMP, DR5, ROS, and Nrf2	Jin et al., 2018 [103]
Invasion ↓	Regulation of EMT and COX-2/MMP2,9/ZEB1, Snail, and miR-200c/ZEB1 pathways	Shan et al., 2013 [93]
Migration ↓	Regulation of autophagy activation	Bao et al., 2014 [112]

IGFBP, insulin-like growth-factor-binding protein; NF-κB, nuclear factor-kappa B; HIF, hypoxia-inducible factor; PARP, poly ADP-ribose polymerase; EGFR, epidermal growth factor receptor; HER, human EGFR-related; ROS, reactive oxygen species; ER, endoplasmic reticulum; Nrf2, nuclear factor erythroid 2-related factor; MMP, matrix metalloproteinase; DR5, death receptor 5; EMT, epithelial-to-mesenchymal transition; COX, cyclooxygenase.

**Table 4 molecules-25-00575-t004:** *In vitro* molecular mechanism of the anti-cancer effects of phenethyl isothiocyanate.

Anti-Cancer Effect	Underlying Molecular Mechanisms	Reference
Apoptosis ↑	Via caspase-9, a major regulator, and Bcl-2, Bax, Bak, and Bcl-xl	Tang & Zhang, 2005 [70]
Apoptosis ↑	Decreased NF-κB, survivin, Twist, and Akt expression and increased PTEN and JNK expression	Tang et al., 2013 [123]

Bcl, B-cell lymphoma-2; Bax, Bcl-2-associated X protein; Bak, BCL2-antagonist/killer; NF-κB, nuclear factor-kappa B; JNK, c-Jun N-terminal kinase.

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
