# Peer review of "Molecular Mechanisms of the Anti-Cancer Effects of Isothiocyanates from Cruciferous Vegetables in Bladder Cancer"

_molecules, 2020, doi:10.3390/molecules25030575_

Round 1
Reviewer 1 Report
Bladder cancer is one of the cancers of the genitourinary system. It occurs in men almost three times more often than in women. This tumor is most often caused by the transitional epithelium lining the bladder and may take the form of a papilloma (papilloma / Latin papilloma) that penetrates inside the bladder.
I recommend the authors to describe more broadly the factors increasing the risk of developing bladder cancer:
We do not know the exact mechanism leading to the change of the normal bladder epithelial cell into cancer. It is only known that mutations and instability of the cell genome contribute to this.
For many years, scientists have been trying to create a list of factors that increase the risk of developing bladder cancer. - It is worth to prepare such a list
After all, it is well known that smokers are four times more likely to have cancer of the genitourinary system.
4-aminobiphenyl belonging to aromatic amines is a carcinogen present in cigarette smoke as well as in many industrial chemicals. The formation of carcinogenic compounds is mediated by liver proteins such as cytochrome P450, N-acetyltransferase and S-glutonion transferase. Cytochrome P450 catalyzes, among others hydroxylation of aromatic amines, which get into the bladder epithelium, are substrates for N-acetyltransferase1. The result of this reaction is the formation of compounds that cause DNA mutations. Hydroxylamine may also be affected by hepatic S-glutathione transferase, which contributes to the formation of aromatic amines that are not destructive to DNA. Chronic cystitis is also a factor that increases the risk of cancer.
The mechanism of anti-carcinogenicity of isothiocyanates is:
- ability to induce the expression of phase II enzymes;
- inhibition of phase I enzymes responsible for the metabolic activation of carcinogens;
- blocking DNA damaging factors;
- limiting the transformation of initiated cells and restoring apoptosis, i.e. programmed cancer cell death;
- participation in estrogen metabolism
It is worth mentioning about this scientific information!
Briefly describe the mechanism of bladder cancer formation and treatment. Focus on the mechanism itself.
Bladder cancer is three times more common in men, are they more likely to carcinogenic?
I did not have enough information on the effects of testosterone
The male hormone testosterone increases cell division, but does not affect the programmed death pathway (apoptosis). The intensity of cell division is after all dependent on the concentration of testosterone in the body. The higher the dose of this hormone, the higher the number of proliferating cells.
Describe isothiocyanates and their origin, e.g. from various food products, usually their mechanisms
The first group includes isothiocyanates such as phenethyl isothiocyanate (PEITC) and sulforaphane of greatest importance in research on
chemoprevention. Mechanism of action of isothiocyanates is based on the disruption of the binding between Nrf2 and the Keap 1 protein, the consequence of which is dissociation Nrf2 from an inactive cytoplasmic complex. The release of Nrf2 from Keap1 is also possible due to the activation of protein kinases (MAPK, PKC, PI3K, PERK). IN as a consequence, transcription of containing genes increases ARE element. Organic isothiocyanates, commonly found in the human diet, mainly in mustard and horseradish, are therefore very strong GST inducers. and NQO1. It has been proven that in this way PEITC inhibits various forms of animal induced cancers experimental by chemical carcinogens such like nitrosamines. Sulforaphane is insulated with broccoli one of the strongest inducers of GST and NQO1. It has been shown, among others, that this compound inhibits the development of cancer rat mammary gland. Similarly, through induction of phase II enzymes, mainly GST and NQO1, said synthetic compound, oltipraze, works.
In conclusion I can write that:
Bladder cancer, as already mentioned, is one of the more frequently occurring disorders of the genitourinary system, however, many years of research and early detection of disturbing changes have significantly reduced the mortality of patients. Understanding the factors that increase your risk of getting sick helps protect against bladder cancer. Although the treatment of cancer is very difficult, learning the mechanisms of cancer formation and the effects of drugs allows for a more effective fight against this disease. The use of gene therapy in the treatment of cancer creates great opportunities. Treating patients through genetic manipulation of cells would allow the mutated gene to be repaired and the causes of the disease eliminated. The possibilities of gene therapy are the subject of research of many scientists, but its use in hereditary or acquired diseases is not fully understood.
Congratulations to the authors of a very interesting article.
Author Response
We thank the reviewers for carefully evaluating our manuscript. The suggestions and advices have greatly helped us in improving the manuscript. Our point-by-point responses to the comments are provided below. Changes made in response to the comments are highlighted in red in the revised version of the manuscript. In addition, our manuscript has been edited by a professional editing service.
(Comments)
Bladder cancer is one of the cancers of the genitourinary system. It occurs in men almost three times more often than in women. This tumor is most often caused by the transitional epithelium lining the bladder and may take the form of a papilloma (papilloma / Latin papilloma) that penetrates inside the bladder.
I recommend the authors to describe more broadly the factors increasing the risk of developing bladder cancer:
We do not know the exact mechanism leading to the change of the normal bladder epithelial cell into cancer. It is only known that mutations and instability of the cell genome contribute to this.
For many years, scientists have been trying to create a list of factors that increase the risk of developing bladder cancer. - It is worth to prepare such a list
After all, it is well known that smokers are four times more likely to have cancer of the genitourinary system.
4-aminobiphenyl belonging to aromatic amines is a carcinogen present in cigarette smoke as well as in many industrial chemicals. The formation of carcinogenic compounds is mediated by liver proteins such as cytochrome P450, N-acetyltransferase and S-glutonion transferase. Cytochrome P450 catalyzes, among others hydroxylation of aromatic amines, which get into the bladder epithelium, are substrates for N-acetyltransferase1. The result of this reaction is the formation of compounds that cause DNA mutations. Hydroxylamine may also be affected by hepatic S-glutathione transferase, which contributes to the formation of aromatic amines that are not destructive to DNA. Chronic cystitis is also a factor that increases the risk of cancer.
The mechanism of anti-carcinogenicity of isothiocyanates is:
- ability to induce the expression of phase II enzymes;
- inhibition of phase I enzymes responsible for the metabolic activation of carcinogens;
- blocking DNA damaging factors;
- limiting the transformation of initiated cells and restoring apoptosis, i.e. programmed cancer cell death;
- participation in estrogen metabolism
It is worth mentioning about this scientific information!
Briefly describe the mechanism of bladder cancer formation and treatment. Focus on the mechanism itself.
Bladder cancer is three times more common in men, are they more likely to carcinogenic?
I did not have enough information on the effects of testosterone
The male hormone testosterone increases cell division, but does not affect the programmed death pathway (apoptosis). The intensity of cell division is after all dependent on the concentration of testosterone in the body. The higher the dose of this hormone, the higher the number of proliferating cells.
Response: Thank you for your insightful suggestion, which we are in agreement with. We have thus included a new section 4 titled “Further considerations” and discussed factors such as smoking, phase II enzymes, estrogen, and testosterone. Accordingly, new references (142-149) have been included in the text.
(Comments)
Describe isothiocyanates and their origin, e.g. from various food products, usually their mechanisms.
The first group includes isothiocyanates such as phenethyl isothiocyanate (PEITC) and sulforaphane of greatest importance in research on
chemoprevention. Mechanism of action of isothiocyanates is based on the disruption of the binding between Nrf2 and the Keap 1 protein, the consequence of which is dissociation Nrf2 from an inactive cytoplasmic complex. The release of Nrf2 from Keap1 is also possible due to the activation of protein kinases (MAPK, PKC, PI3K, PERK). IN as a consequence, transcription of containing genes increases ARE element. Organic isothiocyanates, commonly found in the human diet, mainly in mustard and horseradish, are therefore very strong GST inducers. and NQO1. It has been proven that in this way PEITC inhibits various forms of animal induced cancers experimental by chemical carcinogens such like nitrosamines. Sulforaphane is insulated with broccoli one of the strongest inducers of GST and NQO1. It has been shown, among others, that this compound inhibits the development of cancer rat mammary gland. Similarly, through induction of phase II enzymes, mainly GST and NQO1, said synthetic compound, oltipraze, works.
Response: Thank you for your insightful comment. In the revised manuscript, we have summarized the importance of these factors in section 4 “Further considerations.”
(Comments)
In conclusion I can write that:
Bladder cancer, as already mentioned, is one of the more frequently occurring disorders of the genitourinary system, however, many years of research and early detection of disturbing changes have significantly reduced the mortality of patients. Understanding the factors that increase your risk of getting sick helps protect against bladder cancer. Although the treatment of cancer is very difficult, learning the mechanisms of cancer formation and the effects of drugs allows for a more effective fight against this disease. The use of gene therapy in the treatment of cancer creates great opportunities. Treating patients through genetic manipulation of cells would allow the mutated gene to be repaired and the causes of the disease eliminated. The possibilities of gene therapy are the subject of research of many scientists, but its use in hereditary or acquired diseases is not fully understood.
Congratulations to the authors of a very interesting article.
Response: Thank you for your positive evaluation. We have also added a sentence on the clinical utility of a combination therapy of ITCs and new therapeutic strategies, including gene therapy, for patients with BC in section 5 (Conclusions).
Reviewer 2 Report
Authors reviewed the anti-bladder cancer effects of isothiocyanates from cruciferous vegetables. Well designed but still need some parts to be revised.
1. "3. Allyl isothiocyanate 4. Benzyl isothiocyanate 5. Sulforaphane 6. Phenethyl isothiocyanate" would be better to be subheading in "2. Isothiocyanates in cruciferous vegetables" like 2.1. Allyl isothiocyanate.
2. Please erase the empty row of tables. Unify the form of "Underlying molecular mechanisms" in tables.
3. The forms are different from each other. Are references corrected cited in the tables? The citation no. should be in [].
4. Explain all the abbreviation at their first appearances. Or abbreviation section is required. "TICs" is not explained.
5. Because there are some drug like "Suppression at high doses and promotion at low doses via regulation of autophagy". Dose part is needed in the tables.
Author Response
We thank the reviewers for carefully evaluating our manuscript. The suggestions and advices have greatly helped us in improving the manuscript. Our point-by-point responses to the comments are provided below. Changes made in response to the comments are highlighted in red in the revised version of the manuscript. In addition, our manuscript has been edited by a professional editing service.
(Comments)
Authors reviewed the anti-bladder cancer effects of isothiocyanates from cruciferous vegetables. Well designed but still need some parts to be revised.
Comment 1. "3. Allyl isothiocyanate 4. Benzyl isothiocyanate 5. Sulforaphane 6. Phenethyl isothiocyanate" would be better to be subheading in "2. Isothiocyanates in cruciferous vegetables" like 2.1. Allyl isothiocyanate.
Response: Thank you for your pertinent suggestion. We have accordingly modified the subheadings.
Comment 2. Please erase the empty row of tables. Unify the form of "Underlying molecular mechanisms" in tables.
Response: We apologize for the discrepancies and have accordingly edited and unified all Tables.
Comment 3. The forms are different from each other. Are references corrected cited in the tables? The citation no. should be in [].
Response: We apologize for these oversights. All citations in the Tables were changed to citations within brackets.
Comment 4. Explain all the abbreviation at their first appearances. Or abbreviation section is required. "TICs" is not explained.
Response: We apologize for these mistakes. “TICs” was a misspelling of “isothiocyanates (ITCs),” which has been accordingly corrected. In addition, some abbreviations were not defined at first mention. this includes BBN, UC, etc. I have corrected these errors.
Comment 5. Because there are some drug like "Suppression at high doses and promotion at low doses via regulation of autophagy". Dose part is needed in the tables.
Response: Thank you for your insightful comment. As we are concerned with the anti-cancer effects of ITCs in this review, we modified Table 3 to just show the molecular mechanisms underlying these anti-cancer effects of sulforaphane.
Round 2
Reviewer 2 Report
Authors revised the manuscript casfully. However, some parts should be modified before publication.
The "underlying molecular mechanisms in table" section could be unified and simplified by using arrows like "anti-cancer effect section". For example, instead of "Increased IGFBP-3 expression and decreased NF-κB expression", ↑IGFBP-3, ↓NF-κB could be used. Author can erase authors name, year in reference of table, if the reference numbers are linked to references. Instead of "Savio et al., 2015 [46]", "[46] can be used.